# Exploring Online Health Information Seeking Behaviour (OHISB) among young adults: A scoping review protocol

Kristine Moksnes Bratland [ID] , Charlotte Wien [ID] , Torkjel M Sandanger

Department of Community Medicine, UiT The Arctic University of Norway, Tromso, Troms og Finnmark, Norway

**Correspondence to**
Kristine Moksnes Bratland;
kristine.m.bratland@uit.no

## ABSTRACT

**Introduction** In recent years, information technology and social media have experienced unprecedented growth, particularly in the Nordic countries. However, there is a noticeable lack of comprehensive understanding regarding the latest research findings on online health information seeking behaviour (OHISB) among young adults (18 to >30). There is a need to conduct an updated review to identify knowledge gaps in where young adults find health information and their user interface preferences and to provide research-based guidance and recommendations to governments, health organisations and social media platforms on how to facilitate this prominent pattern. The scoping review protocol outlines a study that will systematically map the existing literature on young adults' preferences for digital platforms and platform characteristics in relation to OHISB, enabling the identification of promising areas for further research and the development of more effective interventions to promote healthy and informed choices. Conducting a scoping review is imperative to gain a comprehensive understanding of young adults' OHISB and support the next generation of dissemination that promotes accurate and reliable digital health information.

**Methods and analysis** The scoping review will use Arksey and O'Malley's methodological framework (Preferred Reporting Items for Systematic Reviews and Meta-Analyses extension for Scoping Reviews (PRISMA-ScR)) and employ the citation pearl method and the Sample, Phenomenon of Interest, Design, Evaluation, Research type model to design the search strategy. To identify relevant literature, three databases will undergo a search: Scopus, Web of Science and EMBASE. Additionally, a subsidiarily grey literature search will be conducted in Google Scholar. The data charting process will conform to the PRISMA-ScR standard and will be further structured with EndNote. Qualitative and quantitative analyses of the extracted data will be developed using EndNote and Excel.

**Ethics and dissemination** Conducting a scoping review involves secondary data analysis of publicly available sources and does not require an ethical review. The protocol will be published to ensure transparency. The scoping review results will be disseminated through open-access peer-reviewed publications, national and international conferences, social media platforms, newspapers and YouTube to service users and stakeholders.

## STRENGTHS AND LIMITATIONS OF THIS STUDY

⇒ A systematic study was conducted following the Sample, Phenomenon of Interest, Design, Evaluation, Research type model using EndNote and the Preferred Reporting Items for Systematic Reviews and Meta-Analyses extension for Scoping Reviews checklist.
⇒ Limited to include articles published within the last 6 years.
⇒ Conducted in an underserved and rapidly developing cross-disciplinary research field.

## INTRODUCTION

Johnson[12] defined health information-seeking behaviour (HISB) as a deliberate attempt to meet a health information need, distinguishing it from passive forms of information acquisition, such as incidental exposure or scanning.[3] The emergence of online health information seeking behaviour (OHISB) in the late 1990s and early 2000s coincided with the advent of the internet as a promising source of health information.[1 4 5] In the last decade, research on OHISB has grown significantly, driven by the rapid increase in health information available and the shift from an authoritative clinician–patient interaction to shared decision-making.[6]

The definition of young adults varies across contexts and disciplines. In the USA, young adults are typically defined as individuals between the ages of 18 and 34,[7] while in Europe, the age range is often extended to 18–39.[8] Being a young adult represents unique challenges and opportunities,[9] involving the transition from adolescence to adulthood, as well as taking responsibility for their own health. Consequently, information needs, actions and behaviours differ in adolescents, adults and seniors, which reflect their behaviours.[10]

Exploring the information preferences of young adults is important due to their status as *modern* digital native[11] individuals who have grown up with digital and portable

technology. As the first generation to grow up with the internet and related technologies, young adults have distinct OHISB and preferences compared with the older generation.[12] Understanding these differences is crucial for promoting and improving health outcomes among this population.[10 12] Conducting research on young adults' OHISB is essential for identifying gaps in health knowledge, improving health literacy and promoting health equity.[13–15] Young people are more likely to seek out online health information than older adults, and the use of OHISB is increasing among young people.[16 17] However, we lack an overview of the factors driving these outcomes, and it is still important to gain a deeper understanding of their OHISB. This is because young adults may have unique information needs and preferences, and understanding these factors can help healthcare providers and policymakers develop more effective strategies for promoting health literacy and improving health outcomes among this population.

Understanding how young adults use digital tools to seek and process health information and how this information shapes their health behaviours is important. Young adults, being heavy users of technology and digital information,[16 17] make a critical audience for online health information.[18 19] Reviewing the current state of research on OHISB is crucial for informing health communication and intervention strategies due to the potential impact of such information on their health behaviours and outcomes. Such a review is also novel in that it provides a comprehensive synthesis of existing studies, which are often fragmented and lack consistency in the conceptualisation and measurement of key constructs.

This study aims to address the lack of understanding regarding OHISB behaviour among young adults. By conducting an updated scoping review, the study will identify knowledge gaps. This protocol advocates using the scoping review methodology for exploring a broad research topic such as OHISB among young adults.[20 21]

## METHODS AND ANALYSIS

Scooping reviews use a variety of study designs to synthesise evidence and provide a comprehensive summary that informs practice, programmes and policy, while also guiding future research priorities.[20–22] It differs from systematic reviews in its purpose and aims, representing a relatively new approach to synthesising evidence.[23] Both reviews use rigorous and transparent methods to identify and analyse relevant literature.[24] Where systematic reviews aim to sum up the best available research on a specific question from a relatively smaller number of studies pertaining to a focused research question, the purpose of a scoping review is to map the body of literature on a topic or area.[24 25]

The search strategy is based on the scoping review methodology suggested by Arksey and O'Malley[20] and further developed by Levac, Colquhoun and O'Brien[21], and Peters *et al*.[26] The purpose of a scoping review is to

provide an overview of the available research evidence without producing a summary answer to guide clinical decision-making[20 25] and identify gaps in the literature.[21 22]

While the methodical framework of Arksey and O'Malley,[20] further developed by Levac *et al*,[21] informs the overall conduct of the scoping review, the Preferred Reporting Items for Systematic Reviews and Meta-Analyses extension for Scoping Reviews (PRISMA-ScR)[27] guides the reporting of this protocol and also subsequently structures the reporting of the full review.

### Stage 1: defining the research question

This protocol aims to describe a possible, research-based approach on how to carry out a scoping review to get an overview of the body of literature on OHISB among young adults. Furthermore, this protocol does not include a research question, as it is a preliminary plan for conducting a study. However, a research question has been developed for the study and paper that will be conducted based on this protocol. The overall purpose of the planned scoping review presented in this protocol is:

What do previous studies report on young adults' OHISB on health information regarding the choice of digital platform and platform user interface (UI)?

The review is guided by the following objectives:
1. Identifying studies on young adults' OHISB.
2. Determining which digital platforms young adults tend to search for health information.
3. Identifying characteristics in the UI that apply to young adults' aims and trust.
4. Enhancing the characteristics that have an impact.

### Stage 2: identifying relevant literature

In the scoping review on OHISB among young adults, a systematic approach is used to choose the search terms. The following steps are taken: a preliminary search is conducted with the support of subject-matter experts in the field to identify additional keywords and search terms. Controlled vocabulary (MeSH-terms) and keywords are used to build the first search string, which results in the identification of three relevant papers. Citation Pearl Growing[28] is applied by thoroughly reading the three core papers from the preliminary search and searching for new search terms. The bibliography is revisited to look for relevant references published prior to the core documents that could contain other possible search terms. The search string is revisited and developed to ensure that it captures all the relevant concepts related to the research question. The search string is presented to a new group of experts in the field for feedback, and changes are made accordingly. The initial search string is designed using key elements from the review question, applying the Sample, Phenomenon of Interest, Design, Evaluation, Research type (SPIDER) strategy (see table 1). The SPIDER tool is an alternative to the more well-known population, intervention, comparator and outcome (PICO) and is better suited when searching for qualitative research papers[29]

**Table 1** The SPIDER tool applied to the review question

| | |
|---|---|
| Sample | Young adults in the age range where they make their own choices, meaning aged over 18 and under 30 years. No limitations in gender. |
| Phenomenon of Interest | Online Health Information-Seeking Behaviour |
| Design | Published literature of any research design, grey literature |
| Evaluation | 'Searching activities'—characteristics, views, experiences, observations, descriptions, surveys and descriptives. |
| Research type | Qualitative, quantitative, mixed-methods, multi-methods peer-reviewed studies. Grey literature including the third sector and conference proceedings. |

'e', replacing the 'O' (outcome) with the broader term 'E' (evaluation).[29]

The proposed search strategy mapped against the SPIDER tool is presented in table 2.

Boolean operators (AND, OR) are used to combine the various search terms, ensuring that relevant articles that meet the inclusion criteria are retrieved. Multiple rounds of trials and errors will be conducted. Through these techniques, the search strings are ensured to be comprehensive and capture the relevant concepts[30] related to OHISB among young adults.

When conducting a scoping review, it is important to choose appropriate databases to ensure that all relevant studies are captured.[20 26] To identify relevant articles for the scoping review on OHISB among young adults, a comprehensive search is conducted on three electronic databases: EMBASE, Web of Science and SCOPUS. These three databases are commonly used in health research, and each has its own strengths and weaknesses.[31 32] By searching multiple databases, the risk of missing important studies that may be indexed in one database but not in another is reduced. These databases also have advanced search capabilities that allow for refinement of the search strategy and retrieval of only the most relevant studies.[32] They are widely used in health research, which means that the review can be compared and contextualised with other studies that have used similar methods and data sources.

After identifying the relevant articles from the databases, the citation pearl method will be used to identify additional relevant studies. This method involves examining the reference lists of the selected articles to identify any other relevant studies that may not have been captured in the initial search.[33] A forward citation search will also be conducted to identify any new articles that cite the selected articles since their publication. This will help to identify newer studies that may not have been captured in the initial search or the citation pearl search. The citation pearl method is a commonly used technique in systematic reviews and scoping reviews to ensure that all relevant studies are included in the analysis.[27] It is particularly useful when searching for grey literature or when there are no standard indexing terms to identify relevant articles. By using this method, additional studies that meet the inclusion criteria will be identified, ensuring that the scoping review is comprehensive and robust. After the advanced search, a search will be conducted in Google Scholar using the advanced search feature to detect grey literature. Specific search filters will be applied, such as limiting results to conference proceedings, reports, theses, dissertations and other non-peer-reviewed sources. Furthermore, an advanced search will be conducted for grey literature using Google Scholar to ensure that no relevant studies are missed.[30] By using Google Scholar, our search for grey literature will be broadened beyond traditional academic sources to identify additional studies on OHISB among young adults. A combination of keywords and advanced search techniques will be employed to retrieve relevant grey literature.[33] This approach will help identify a range of studies that may have been missed in the initial database search. By using traditional academic databases and Google Scholar to search for grey literature, the scoping review on OHISB among young adults will be made as comprehensive as possible, including all available evidence on the topic.

### Stage 3: article selection

To identify papers about OHISB in young people, a set of inclusion and exclusion criteria is developed. The aspects are presented in table 3. The following will be the criteria for inclusion:
1. The topic of the article must be health-related or encompassing topics, for example, mental health, diet or nutrition.
2. The study must target young adults, specifically address the population of young adults (aged 18–30 years) or make explicit, equivalent assertions. Although the search strategies indicate two commonly accepted lower age boundaries, 'early 20s' and 'aged 20' to identify

**Table 2** The proposed search strategy mapped against the SPIDER tool

| | |
|---|---|
| Sample | ('young adults' OR young OR 'early adulthood' OR 'early 20s' OR 'aged 20') AND |
| Phenomenon of Interest | 'Health Information' AND |
| Evaluation | AND (search* OR seek* OR find* OR access* OR retrieve* OR behaviour*) AND (internet OR online OR web OR digital* OR media) |

**Table 3**  Inclusion and exclusion criteria

| Criterion | Definition |
|---|---|
| Population | Population of young adults (>18 and ≤30), or explicit address this part of the population. Surveys that also screen children or adults will be included if the procedure for young adults is described. |
| Screening procedure | Peer-reviewed publications on a health-related topic with the aim of describing, elaborate, systematise and/or quantifying OHISB, use of the internet, search engines, browsers, web 2.0 or 3.0, social media or other online platforms retrieving information. |
| OHISB definitions | A qualitative or quantitative description of an active effort to obtain health-related information about a specific topic, using any digital sources to obtain the desired information, that occurs outside of routine patterns of interpersonal communication or media use.[6 18 36] |
| Platform | Interested in determining what digital platforms young adults tend to use when searching for health information. |
| Characteristics | Interested in identifying traits, characteristics or interfaces on these digital platforms that young adults prefer, enjoy or have a significant meaning for those using the exact platform. |
| Setting | Primarily observational studies or self-reported studies from the global north. However, it may include studies developed in other parts of the world if the methodology employed is subsequently. |

OHIS, online health information seeking; OHISB, online health information seeking behaviour.

young adults above teens, it will not exclude other ways to describe the population.

3. Describe OHISB activities or individual preferences or assessments, for example, general OHIS, search strategies, choice of health information sources or use of health information.
4. The paper must be published in 2018 or later to make the results applicable to today's digital landscape.

The exclusion criteria are as follows:

1. Studies that focus on the use of general information and communication technology or behaviour change instead of health information will be excluded.
2. Papers where age is a determinant in studying the OHIS of the overall population will be excluded, as it is apparent that age has an influence on individual OHISB. Abstracts, posters, op-eds or letters will be excluded as they are not full papers.
3. Papers that are not published in a peer-reviewed journal or in the proceedings of a conference will be excluded.
4. Papers written in languages other than English, Norwegian, Swedish or Danish will be excluded.

When the search is conducted and the articles are retrieved in EndNote, the next step will involve screening them for eligibility based on established criteria. A three-stage screening process will be performed, where researchers will first remove duplicates. The next step will be to screen titles and abstracts to exclude obviously irrelevant articles. This will be followed by a full-text screening of the remaining articles to determine whether they meet the inclusion criteria. Additionally, EndNote will be used as citation management software to facilitate the screening process and keep track of the articles throughout the review. Each step of the process will be filed in separate EndNote libraries. The reasons for excluding articles will be documented, and the progress will be made visible by presenting a PRISMA flow diagram, which will include searches of the databases, registers and other sources. This will ensure the transparency and reproducibility of the review process.

Once the search is conducted and the articles are retrieved and stored in a labelled EndNote library, the next step will involve screening them for eligibility based on the established criteria. This will involve a two-stage screening process, where titles and abstracts will be first screened to exclude obviously irrelevant articles, followed by a full-text screening of the remaining articles to determine whether they meet the inclusion criteria. The results from each step will be stored in newly separated EndNote libraries. It will be important to document the process of excluding articles to ensure transparency and reproducibility of the review process and to make the scoping review comprehensive, relevant and reliable.

### Stage 4: data charting

To record and synthesise the extracted data, a data charting form will be used. The suggested form is presented in table 4. The form will help to organise the data into categories and subcategories and identify gaps in the literature. Descriptive statistics such as frequency and percentage will be used to summarise the extracted data. The findings of the review will help to identify current trends, patterns and factors influencing young adults' OHISB, including the benefits and risks associated with OHISB. The results will also inform the development of strategies to improve the quality and accessibility of online health information for young adults.

### Stage 5: collating, summarising and reporting the results

EndNote and Excel will be used to arrange and scrutinise the data extracted from the full-text articles. The goal is to produce both quantitative and qualitative data while analysing the articles comprehensively to evaluate them and achieve the objectives of the review. Excel will

**Table 4** Variables and definitions for data chartering

| Variable | Definition |
|---|---|
| First author | The last name of the paper's first author |
| Year | The year the paper was published |
| Title | Title of the article |
| Journal | Name of the journal |
| Nationality | Nationality of the publisher or university |
| Country | Country or nationalities of the population |
| Sample size | The number of participants, informants or observations |
| Population | Group, sex and age |
| Method | Types of data and/or tools: qualitative (survey, interview, administrative data, observation and focus group) and quantitative (check list, self-report, chart reviews, electronic health record and validated questionnaire) |
| Results | The main results or findings |
| Other notes | Additional information about the paper |

be used to analyse the data generated, and following the PRISMA guidelines,[27] assessments of the qualitative and quantitative data will be integrated to present the results. Findings will be tabulated by identifying:

1. Where young adults report finding health information.
2. Choice of digital platform.
3. Preferences regarding UI.

### Patient and public involvement

This study is a scoping review, which involves searching for and mapping the existing literature on a particular topic. As such, there will be no direct involvement of the public in the study beyond the dissemination and publication of the results.

### Expected limitations and strengths of review findings

The upcoming scoping review on OHISB among young adults will aim to identify research gaps, map synthesise evidence and summarise findings using a transparent and systematic approach. The study will focus on identifying which digital platforms young adults tend to use to search for health information, as well as the characteristics of the UI that young adults trust and find useful when seeking health information online. The scoping review will be conducted using the Arksey and O'Malley[20] framework and the PICO model[34] to develop a search strategy, while Excel will be used to chart and organise the data, and the PRISMA-ScR checklist[27] will guide the reporting of findings. To ensure the quality of the scoping review, the search will be limited to studies published in peer-reviewed journals. However, the heterogeneity of the included studies will be acknowledged to limit the ability to provide a quantitative synthesis of the data, such as a meta-analysis. Instead, a focus will be placed on a qualitative synthesis of the evidence, which will provide

a comprehensive overview of the research in the field. By synthesising the evidence, gaps in the literature will be identified and insights into the characteristics of digital platforms that are most effective for young adults when seeking health information online will be provided. The findings will be reported in a peer-reviewed journal, following the scoping review reporting standards (PRISMA-ScR) and disseminated widely to stakeholder groups using innovative media approaches.

## ETHICS AND DISSEMINATION

Ethical considerations will remain essential in all health research, including scoping reviews. In Norway, the ethical principles for medical research are set out in the Norwegian Health Research Act,[35] which emphasises the importance of protecting the rights and privacy of research participants. As this scoping review will not involve human research participants, it will not be necessary to obtain informed consent, protect anonymity or receive institutional board approval. However, we will still ensure the integrity of the research and respect for all potential stakeholders by following established guidelines for conducting scoping reviews, such as the PRISMA-ScR checklist[27] and the Arksey and O'Malley[20] framework. The findings will be disseminated through various channels, including open-access peer-reviewed journal publications, national and international conference presentations, academic social media platforms, newspaper articles and blog posts. By sharing our results widely, the aim will be to inform and engage a diverse audience, including individuals affiliated with our institutions and professional associations.

**Contributors** This protocol was developed by KMB. CW and TMS have contributed to the study's conception and design. KMB developed the search string in collaboration with two senior librarians from UiT and SDU. The search string presented has been discussed between KMB, CW and TMS during the process. Conducting the final retrieving of literature, KMB will prepare the dataset and conduct the analysis. During the analysis, the findings will be presented by KMB and put up for debate between KMB, CW and TMS. KMB will then write the first draft of the article manuscript, which will be critically reviewed and refined by CW. Both CW and TMS will comment on the following versions of the manuscript. KMB, CW and TMS will read and approve the final version of the manuscript before submission.

**Funding** The PhD project is funded by UiT, the Arctic University of Norway and the Healthy Choices and Social Gradient project, funded by the Research Council of Norway (grant number 289440).

**Competing interests** None declared.

**Patient and public involvement** Patients and/or the public were not involved in the design, or conduct, or reporting, or dissemination plans of this research.

**Patient consent for publication** Not applicable.

**Provenance and peer review** Not commissioned; externally peer reviewed.

**ORCID iDs**
Kristine Moksnes Bratland http://orcid.org/0000-0002-7611-056X
Charlotte Wien http://orcid.org/0000-0002-3257-2084

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
