## [Reviewer comments · BMJ Open]

ARTICLE DETAILS

TITLE (PROVISIONAL)	Exploring online health information seeking behavior (OHIS) among young adults. A Scoping Review Protocol
AUTHORS	Bratland, Kristine; Wien, Charlotte; Sandanger, Torkjel

VERSION 1 – REVIEW

REVIEWER	Navarro, Jessica Elon University
REVIEW RETURNED	27-Jul-2023

GENERAL COMMENTS	The study protocol presented for the scoping review on health information seeking behavior among young adults is a promising research endeavor that holds significant potential in understanding how youth access and navigate health-related information online. The proposed research aims to shed light on the growing reliance on digital platforms for health-related inquiries. As a reviewer, I find this study to be of great interest, and I eagerly anticipate reading the final results. The study protocol demonstrates a comprehensive approach, and I have two suggestions that the authors may consider for further improvement. Firstly, I noticed that the search strategy primarily focuses on the term "health information" as the phenomenon of interest. While this is a relevant and broad starting point, it might inadvertently exclude valuable studies that use more specific terms related to health issues, such as "HIV/AIDS knowledge" or "depression" or other mental health diagnoses. To ensure inclusivity, I recommend expanding the search strategy to incorporate a wider range of relevant keywords. This adjustment could capture studies that explore health information seeking behavior in the context of various specific health topics, leading to a more comprehensive and diverse review. Secondly, in relation to the findings, I believe it would be extremely valuable to include data on how reputable young adults perceive the reliability of health information sources they encounter online and how confident they feel in navigating the vast array of information available. Digital literacy is a critical aspect of information-seeking behavior and understanding how young adults discern credible sources and make informed decisions regarding their health can offer significant insights. By incorporating this dimension into the study's scope, the authors can enrich their findings and provide a more nuanced perspective on health information seeking behavior among young adults.
---

	I look forward to reading the final results and believe that this study could serve as a valuable resource for scholars, practitioners, and policymakers alike.
--	---

REVIEWER	McMullan, Ryan Macquarie University Faculty of Medicine and Health Sciences, Australian Institute of Health Innovation
REVIEW RETURNED	26-Sep-2023

GENERAL COMMENTS	The authors propose a scoping review of literature that examines online health information seeking in young adults. However, why this scoping review is important and necessary is unclear for two reasons 1) there have been at least two systematic reviews on online health information seeking published since 2021 (the authors do not cite these papers), and 2) the introduction and conclusions do not emphasise clearly the importance and implications of the work. To be considered for publication, the authors would need to complete major revisions in which they would need to clearly explain how their review updates and provides a better synthesis of the literature on online health information seeking in young adults compared to previous systematic reviews. The authors also need to strengthen the methods and conclusion sections. I have included suggestions for major revisions. There are also grammatical and typographical errors throughout the manuscript. Title Having read the paper it is unclear what the “social gradient WP5 protocol” refers to. It should also be made clear that it is a scoping review study protocol. Abstract In the abstract it is stated that “there is a noticeable lack of comprehensive understanding of the latest research findings on online health information seeking behaviour”. However, there has been a systematic review and meta-analysis (1) and systematic review (2) on this area published in the last 2 years: 1) Xiaohui Wang, Jingyuan Shi & Hanxiao Kong (2021) Online Health Information Seeking: A Review and Meta-Analysis, Health Communication, 36:10, 1163-1175, DOI: 10.1080/10410236.2020.1748829 2) Jia X, Pang Y, Liu LS. Online Health Information Seeking Behavior: A Systematic Review. Healthcare. 2021; 9(12):1740. https://doi.org/10.3390/healthcare9121740 Neither of these papers have been cited by the authors. According to the authors there is importance in providing a comprehensive understanding of young adults’ OHIS, however, they will need to strengthen their argument with reference to the above papers. i.e., they will need to further distinguish their review from these published papers. Introduction Did the previous systematic reviews not synthesise findings from studies that included young adults? The authors do not define “young adult”. What is a young adult? i.e., how do you distinguish between young adults and old adults? Why is this important? The argument for the completion of a scoping review, rather than a systematic review needs to be strengthened.
--

	The introduction is very brief. There is one very broad paragraph on online health information seeking and then two paragraphs that describe why it is important to research this area and why a scoping review is important. However, there is no exploration of online health information seeking i.e., prevalence globally, concepts, theories etc. Discussion of digital platforms and digital user interface would also benefit the reader. Methods Please edit the research question for clarity. The methods would be clearer if written with a more active voice i.e., rather than writing “a preliminary search is conducted...” consider writing “a preliminary search will be conducted...” and so on for the entire method. A description of each database (EMBASE, Web of Science, SCOPUS) is not necessary. A definition of grey literature is not necessary. For Stage 3, the first paragraph can be deleted. The second inclusion criterion is confusing. It is unclear as to what is meant by the description of “early twenties” and “aged 20”. Can you please clarify? The final two paragraphs of the Stage 3 section repeat information e.g., the process of title and abstract screening, full text screening. Expected limitations and strengths of review finding: In this section the authors need to emphasise the implications of the review. How will the review inform strategies to increase the accuracy and reliability of online health information? Be specific.
--	---

VERSION 1 – AUTHOR RESPONSE

Reviewer 1

Nr.	Comments	Reply and changes
1	I recommend expanding the search strategy to incorporate a wider range of relevant keywords. This adjustment could capture studies that explore health information seeking behavior in the context of various specific health topics, leading to a more comprehensive and diverse review.	Thank you for insightful and engaging comment. Your thoughtful feedback has been invaluable in improving the quality and clarity of our work, and we appreciate your expertise and attention to detail. Search strategy: We agree that expanding the search strategy to incorporate a wider range of relevant keywords could capture studies that explore health information seeking behavior in the context of various specific health topics, leading to a more comprehensive and diverse review. While our initial search strategy focused on the term "health information" as the phenomenon of interest, we recognize that this approach may inadvertently exclude valuable studies that use more specific terms related to health issues. We tried out a search strategy including a wider range of relevant keywords regarding attributes, (health, mental health, medicine, drug, nutrition, diet, wellness, illness, symptoms), to ensure inclusivity. During the process of designing the search string, we consulted with two experienced librarians from Norway and Denmark to ensure the accuracy and relevance of the search string and results. Additionally, we revised the

		manuscript with the assistance of a research librarian who has extensive experience in systematic reviews to address any issues. After conducting several tests on different search string designs and consulting with experts, we have found that incorporating a broader range of relevant keywords often leads to an increase in "noise" and irrelevant results. We tested the mentioned attribute keywords, screened abstracts, and did not identify additional relevant terms or keywords. We appreciate your input, unfortunately we believe that incorporating a wider range of relevant keywords into our search strategy will reduce the quality and comprehensiveness of our study. This issue was also addressed in one of our responses to comments from Reviewer 2 (Reviewer 2, section 10), where we provided an extensive answer that included two examples of search strings that we tested to explore the concept. We appreciate the opportunity to clarify our approach and methodology and hope that this information clarifies our study and protocol.
--	--	---

2	Secondly, in relation to the findings, I believe it would be extremely valuable to include data on how reputable young adults perceive the reliability of health information sources they encounter online and	Thank you for your suggestion regarding the inclusion of data on how reputable young adults perceive the reliability of health information sources they encounter online. While we agree that digital literacy is a critical aspect of information-seeking behavior, we have decided not to include this dimension in our current study for several reasons. Firstly, the reliability of health information sources can vary widely depending on the specific health message being conveyed. Therefore, it would be difficult to assess young adults' perceptions of the reliability of health information sources in a comprehensive and meaningful way within the scope of our current study. Secondly, our study aims to provide a broad overview of health information seeking behavior among young
	how confident they feel in navigating the vast array of information available. Digital literacy is a critical aspect of information-seeking behavior and understanding how young adults discern credible sources and make informed decisions regarding their health can offer significant insights. By incorporating this dimension into the study's scope, the authors can enrich their findings and provide a more nuanced perspective on health information seeking behavior among young adults.	adults, and we have limited resources and time to conduct an in-depth analysis of digital literacy. Nonetheless, we acknowledge the value of understanding how young adults discern credible sources and make informed decisions regarding their health. Third, this aspect will be explored in a separate paper (of Bratland's Ph.D.-project), including analysis of longitudinal data from a cohort study, which will allow us to examine levels of health literacy and their confidence in navigating information over time. While we have decided not to include this dimension in our current study, we recognize the importance of digital literacy in health information seeking behavior among young adults and plan to explore it in more detail in a future study. We appreciate your suggestion and believe that this aspect is crucial to understanding young adults' health information seeking behavior. We hope that our future research will provide valuable insights into this topic. Overall, we appreciate your suggestion and will take it into consideration for future research.

Reviewer 2

Nr.	Comments	Reply and changes
-----	----------	-------------------

1	1) two systematic reviews on online health information seeking 2) the introduction and conclusions do not emphasize clearly the importance and implications of the work. 3) a) clearly explain how their review updates and provides a better synthesis of the literature on online health information seeking in young adults compared to previous systematic reviews. 4) b) strengthen the methods and conclusion sections.	Manuscript ID bmjopen-2023-074894. We would like to express our sincere appreciation for the valuable feedback and suggestions provided by Reviewer 2. We have thoroughly reviewed the issues raised and have taken them into careful consideration. To address each of the points raised by the reviewer, we have systematically written a detailed response in the table provided, along with the measures we have taken to improve the manuscript. We recognize the importance of constructive criticism in the peer-review process and are grateful for the opportunity to improve our work. Once again, we extend our gratitude to Reviewer 2 for their insightful comments and suggestions.
2	Title: Having read the paper it is unclear what the “social gradient WP5 protocol” refers to. It should also be made clear that it is a scoping review study protocol.	In response to the concerns raised regarding “social gradient”, we have removed confusing elements in the title. “WP5” address the specific work package of the protocol, and study later on, is associated to. To clarify the title, we have removed both elements.

		Furthermore, we have modified the title of the paper to explicitly indicate that it is a protocol for a scoping review study. These changes were made:  - Title: Page 1, line 1 and line 3. - Ethics and dissemination: Page 2, line 44 and 45 - Methods and analysis: Page 5, line 140 - 146 - Methods and analysis: Page 5, line 146
--	--	---

3 Abstract: In the abstract it is stated that “there is a noticeable lack of comprehensive understanding of the latest research findings on online health information seeking behaviour”. However, there has been a systematic review and meta-analysis (1) and systematic review (2) on this area published in the last 2 years: 1) Xiaohui Wang, Jingyuan Shi & Hanxiao Kong (2021) Online Health Information Seeking: A Review and Meta-Analysis, Health Communication, 36:10, 1163-1175, DOI: 10.1080/10410236.2020.1748829	We have chosen to divide our answer into two parts based upon this response. 1. Phrasing – abstract Thank you for bringing this to our attention. We appreciate the feedback and express our understanding that there may be a need for greater clarity in our work. In the abstract it is stated that “[..]there is a noticeable lack of comprehensive understanding of the latest research findings on online health information seeking behavior (OHIS), particularly among young adults”. Given that this protocol examines a specific cohort, we find the last part of the sentence to have important meaning. We acknowledge that the meaning was not clear enough in the first version of the protocol. We will also address this in our response to the next point in the feedback from reviewer 2.
---	---

2) Jia X, Pang Y, Liu LS. Online Health Information Seeking Behavior: A Systematic Review. Healthcare. 2021; 9(12):1740. https://doi.org/10.3390/healthcare9121740 Neither of these papers have been cited by the authors. According to the authors there is importance in providing a comprehensive understanding of young adults' OHIS, however, they will need to strengthen their argument with reference to the above papers. i.e., they will need to further distinguish their review from these published papers.	These changes were made:  - Abstract: Page 1, line 18 - 20 - Abstract: Page 1, line 22 - Abstract: Page 1, line 29 - 33 - Introduction, page 2 - 3, line 67 – 70 - Introduction: Page 3, line 71 - 77 - Introduction: Page 3, line 78 – 86 - Introduction: Page 4, line 111 - References: Jia et. al., 2021, page 3, line 81, page 3, line 89, and page 18, line 533 – 534 - References: Wang et. al., 2001, page 3, line 81; page 3, line 89, and page 18, line 531 – 532 2. Previous systematic review and meta-analysis (1) and systematic review(2) We would like to thank reviewer 2 for bringing up the two reviews by Wang et al. (2021) and Jia et al. (2021). We are familiar with the reviews and will include these in the introduction of the revised manuscript. However, neither of the reviews do answer to the outlined study described in the proposed protocol, Firstly, there is a lack of a clear sample or cohort in these reviews. Hence, the reviews do not cover the research question described in our protocol that we wish to explore in our paper. Secondly, we miss a more precise presentation of inclusion and exclusion criteria. The reviews would not be possible to include as a part of our scoping review.
---	--

		The reviews cover the general population, with young adults being part of the sample rather than a specifically singled-out cohort. Consequently, the behavior patterns described by the authors as characteristic are “diluted” as the reviews mix results from different age cohorts, including children, teenagers, young adults, adults, and seniors, not singling out behavior according to age. Both reviews acknowledge that there is a correlation between age and behavior, indicating that young people differ from seniors in their actions, see section 3 (The distinction..). The results that we can clearly link to young adults is the increased use of the internet and the tendency to search for more health information as a result (Wang et al., 2021; Jia et al., 2021). This is also a feature that is prominent among women and highly educated individuals (Wang et al., 2021; Jia et al., 2021). The sample in these reviews is not described sufficiently to allow for the separation of individual findings. A more precise presentation of inclusion and exclusion criteria could have made this separation possible. In line with our inclusion criteria, neither of the two studies qualifies to be included in the review, as it contradicts the methodological basis described in the protocol. While this may be considered a limitation, it is consistent with the methodology outlined on page 4 - 9 in the protocol. To provide clarity, the reviews have been included in the manuscript. These changes were made:  - Abstract: Page 1, line 30 – 33 - Introduction: Page 2-3, line 65 – 70 - Introduction: Page 3, line 71 - 77 - Introduction: Page 3, line 81 - 86
--	--	--

		 - Introduction: Page 3, line 89 and 90 - Methods and analysis: Page 4, line 117 – 121 - Methods and analysis: Page 4, line 125 – 127 - References: Page 18, line 531 - 532 - References: Page 18, line 533 - 534 - References: Kim & Xie, 2017, page 3, line 74 and 75 and page 18, line 521 – 522 - References: Prensky 2001, page 3, line 72, and page 18, line 519 – 520 3. The distinction between the aforementioned reviews and our proposed review. Our proposed study described in the protocol stands out from previous literature reviews as it specifically examines the online health information seeking behavior of young adults. It is widely recognized that the digital search behavior of young adults, including the extent, strategies, and sources of information they consult and trust (Kim 2016; Kim & Xie, 2017; Farrugia et al., 2021), differs from that of both younger and older age groups (Miller & Bell, 2012; Farrugia et al., 2021; Zhao, 2022). This demographic possesses unique characteristics as they are the first generation to have had access to the internet and portable technology from birth. Consequently, their behavior is distinct, and their unique features may be overlooked in studies that focus on the entire population. The pattern require facilitation (Jia et. al., 2021). The investigation of where young adults obtain health information, their platform preferences, and the impact of user interface on their behavior, is a novel area of research. While previous studies have identified dominant predictors of OHIS, such as quality, trustworthiness, and usefulness, there
--	--	---

		is a lack of understanding of the practical aspects of OHIS behavior (Wang, 2021). The protocol presents the outline of a study that can provide valuable insights into the practical aspects by examining where young adults are most likely to seek health information that they consider trustworthy and what platforms they prefer. This study can provide a more comprehensive understanding of the digital divide and how it affects young adults' ability to access and comprehend health information online, which can help to address the barriers they face and ensure equal access to reliable and trustworthy health information. Gaining a deeper understanding of young adults is of utmost importance, as interventions aimed at promoting public health must be targeted towards them (Banas, 2008; Kim & Xie, 2017). Additionally, the dissemination of health information must be tailored to their needs (Nguyen et al., 2019; Lim, 2022). Young adults exhibit a preference for accessing information through digital (social) media and are unlikely to revert to traditional media (Lim, 2022). It is crucial to take this change into consideration. These changes were made:  - Abstract: Page 1, line 19 – 20 - Abstract: Page 1, line 29 – 30 - Introduction: Page 2-3, line 67 – 70 - Introduction: Page 3, line 71 – 77 - Introduction: Page 3, line 83 - 86 - Introduction: Page 3, line 88 – 90
--	--	---

- Introduction: Page 4, line 109 – 111
- Methods and analysis: Page 4, 119 – 122
- Methods and analysis: Page 4, 125 – 127
- References: Kim & Xie, 2017, page 3, line 74 and 75 and page 18, line 521 – 522

References:

Banas, J. (2008) A Tailored Approach to Identifying and Addressing College Students' Online Health Information Literacy, American Journal of Health Education, 39:4, 228-236, DOI: [10.1080/19325037.2008.10599043](https://doi.org/10.1080/19325037.2008.10599043)

Farrugia A, Waling A, Pienaar K, Fraser S. The "Be All and End All"? Young People, Online Sexual Health Information, Science and Skepticism. Qualitative Health Research. 2021;31(11):2097-2110. doi:10.1177/10497323211003543 10.1080/10410236.2020.1748829

Jia X, Pang Y, Liu LS. Online Health Information Seeking Behavior: A Systematic Review. Healthcare. 2021; 9(12):1740. <https://doi.org/10.3390/healthcare9121740>

Kim Y. Trust in health information websites: A systematic literature review on the antecedents of trust. Health Informatics Journal. 2016;22(2):355-369. doi:10.1177/1460458214559432

Kim H, Xie B. Health literacy in the eHealth era: A systematic review of the literature. Patient Educ Couns. 2017 Jun;100(6):1073-1082. doi: 10.1016/j.pec.2017.01.015. Epub 2017 Jan 28. PMID: 28174067.

Lim M, Molenaar A, Brennan L, Reid M, McCaffrey T Young Adults' Use of Different Social Media Platforms for Health Information: Insights From Web-Based Conversations J Med Internet Res 2022;24(1):e23656 URL: <https://www.jmir.org/2022/1/e23656> DOI: 10.2196/23656

Miller, L. M., & Bell, R. A. (2012). Online health information seeking: The influence of age, information trustworthiness, and search challenges. Journal of Aging and Health, 24(3), 525–541. <https://doi.org/10.1177/0898264311428167>

Nguyen M, Smets E, Bol N, Loos E, van Laarhoven H, Geijsen D, van Berge Henegouwen M, Tytgat K, van Weert J Tailored Web-Based Information for Younger and Older Patients with Cancer: Randomized Controlled Trial of a Preparatory Educational Intervention on Patient

		Outcomes J Med Internet Res 2019;21(10):e14407 URL: https://www.jmir.org/2019/10/e14407 DOI: 10.2196/14407 Wang, Y., Min, J., Khuri, J., Li, M., & Liang, C. (2021). Health information seeking behaviors during the COVID-19 pandemic among Chinese young adults. International Journal of Environmental Research and Public Health, 18(2), 663. https://doi.org/10.3390/ijerph18020663 Xiaohui Wang, Jingyuan Shi & Hanxiao Kong (2021) Online Health Information Seeking: A Review and Meta-Analysis, Health Communication, 36:10, 1163-1175, DOI: 10.1080/10410236.2020.1748829
--	--	--

		Zhao YC, Zhao M, Song S. Online Health Information Seeking Behaviors Among Older Adults: Systematic Scoping Review. J Med Internet Res. 2022 Feb 16;24(2):e34790. doi: 10.2196/34790. PMID: 35171099; PMCID: PMC8892316.
--	--	--

4 Introduction: Did the previous systematic reviews not synthesise findings from studies that included young adults? The authors do not define "young adult". What is a young adult? i.e., how do you distinguish between young adults and old adults? Why is this important? The argument for the completion of a scoping review, rather than a systematic review needs to be strengthened. The introduction is very brief. There is one very broad paragraph on online health information seeking and then two paragraphs that describe why it is important to research this area and why a scoping review is important. However, there is no exploration of online health information seeking i.e., prevalence globally, concepts, theories etc. Discussion of digital platforms and digital user	We appreciate the crucial feedback provided by Reviewer 2, and we acknowledge the need to address the concerns raised according to the protocol. 1. Definition – young adults The first question regarding the limitations of the two proposed reviews has been addressed in the previous text. Regarding the next question on the term "young adults," we understand the concern and recognize that it could be made clearer. We refer to the operationalization of the term "young adults" in the development of search terms in the search strategy outlined in Table 1: The SPIDER tool applied to the review question, on page 6. We also refer to Table 3: Inclusion and exclusion criteria on page 9. However, we do acknowledge the need for a definition and have provided necessary information both in abstract and introduction. These changes were made:  - Abstract: Elaboration. Page 1, line 29 – 31 - Introduction: Added age limitations, page 1, line 22 - Introduction: Added a definition, page 3, line 65 - 70. - Methods and analysis: Repeated the description of the cohort, page 9, table 3. - Methods and analysis: Changes in inclusion criteria nr. 2, page 9, line 256 - References: Arnett, 2000, page 2, line 67 – and page 18, line 515-516
--	--

interface would also benefit the reader.	 - References: Eurostat, 2021, page 2, line 67 – and page 18, line 514 - Reference: Gray et. al., 2005, page 3, line 70 and 75; and page 15, line 517 – 518 - References: Social Explorer Census Data (n.d.), page 2, line 66 – and page 18, line 512 2. Scoping review vs. systematic review Furthermore, there was a request for an evaluation of the advantage of a scoping review over a systematic review. A scoping review aims to provide an overview of a potentially large and diverse body of literature related to a broad topic. A systematic review attempts to gather empirical evidence from a relatively smaller number of studies related to a focused research question (Weill Cornell Medicine Library. (n.d.). In our case, a scoping review would better illuminate the research question than a systematic review. We appreciate the feedback and have made improvements to the protocol. These changes were made:  - Title: Page 1, line 3 - Abstract: Page 1, line 33 - 35 - Introduction, page 4, line 108 – 111 - Methods and analysis, page 4, line 115 – 122 - Methods and analysis, page 4, line 125 – 127 - Methods and analysis, page 5, line 144 – 147 - References: Sucharew & Macaluso, 2019, page 4, line 118, and page 18, line 546 – 547 - References: Weill Cornell Medicine Library, n.d., page 4, line 119 and 121, and page 18, line 548
---	---

	3. Concepts and theories Thank you for your feedback. We appreciate your engagement in the research question and understand the request for more exploration of the topic OHIS. This document is a protocol. As a protocol, it serves as a detailed plan for conducting a scoping review and is primarily focused on the methodology of the study. While it may not include a comprehensive discussion of all theories and concepts, we agree that these discussions are an essential component of any research article. Therefore, we will ensure that they are included in the final paper. Once again, we appreciate your input and look forward to incorporating it into our work. These changes were made:  - Title: Page 1, line 1 – 2 - Introduction: Page 2, line 71 – 77 - Introduction: Page 3, line 80 – 86 - Introduction: Page 3, line 88 – 89 - Methods and analysis: Page, 5, line 141 – 144 - Methods and analysis: Page 5, line 147 References: Weill Cornell Medicine Library. (n.d.). Scoping reviews. Retrieved from https://med.cornell.libguides.com/systematicreviews/scopingreviews
--	---

5 Research question: Please edit the research question for clarity.	Thank you for bringing this to our attention. We apologize for any confusion caused. As this document is a protocol for a scoping review, the research question for a protocol is not explicitly stated in the manuscript. The protocol is designed to guide the study and the review that will provide the necessary literature to answer the research question mentioned. The research question stated in the protocol is the question to be explored in the proposed study and paper. We acknowledge that this may not have been clear enough and have made changes to improve clarity. These changes were made:  - Abstract: Page 1, line 29 - 30 - Abstract: Page 1, line 39 - Research question: Page 5, 141 - 144
---	---

6	Methods: The methods would be clearer if written with a more active voice i.e., rather than writing “a preliminary search is conducted...” consider writing “a preliminary search will be conducted...” and so on for the entire method.	The phrasing of the protocol has been revised from passive to an active voice to enhance clarity. These changes were made:  - A revision of the full protocol
7	A description of each database (EMBASE, Web of Science, SCOPUS) is not necessary.	We appreciate the input. We will proceed with the protocol without including a description of each database (EMBASE, Web of Science, SCOPUS) as it is not necessary.

		These changes were made:  - Methods and analysis: Description of database removed, page 6-7, line 184 – 193
8	A definition of grey literature is not necessary.	Thank you for your comment. We will proceed with the protocol without including a definition of grey literature. These changes were made:  - Methods and analysis: Definition of grey literature, page 6, line 214 - 218
9	For Stage 3, the first paragraph can be deleted.	Thank you for your suggestion. I have taken it into consideration and made the necessary changes accordingly. These changes were made:  - Methods and analysis: First paragraph deleted, page 8, line 228-232 - Methods and analysis: Inclusion and exclusion criteria moved above table 3 to provide context, page 8, 234 - 255
10	The second inclusion criterion is confusing. It is unclear as to what is meant by the description of “early twenties” and “aged 20”. Can you please clarify?	Thank you for bringing this matter to our attention. We understand that the wording of the second inclusion criterion may have caused confusion. To clarify, our study will include individuals who are over 18 years old and those who are 30 years old or lower (18-30 y.o.). This age range aligns with the commonly accepted definition of young adults. Our study will include participants or cohorts in their late teens, twenties, and

		those who are 30 years old or younger. The terms we have used are
--	--	--

		intended to refine the search strategy and the keywords used in the literature. To ensure a comprehensive selection of literature that deals with young adults and oral health information systems (OHIS), we will describe the method used to select appropriate terminology: The first step in the selection process to develop a functional search string was an initial limited search of at least two relevant online databases, such as MEDLINE and Scopus. This work was conducted in collaboration with two experienced library researchers who specialized in conducting reviews on healthcare and the medical field. We used common terms and ended up with a vast amount of papers that were not relevant to our study. We reviewed thousands of abstracts in search of broader terms. We also designed several different search strings for EMBASE, with and without the "age limit" for young. As an example, one of our earlier search strings from March 10th aimed to narrow down the body of literature by focusing on nutrition in relation to "health information" and excluding terms associated with age and/or "young": ((Twitter or Instagram or Snapchat or Facebook or YouTube or Discord or Twitch or Reddit or TikTok or WeChat or QQ or WhatsApp or Messenger or Skype or Pinterest or Tumblr or instant information).mp. or exp information seeking/ or media literacy/ or social media/ or exp social network/ or social media.mp. or social network*.mp. or ((media or network*) adj5 (communication or "use" or exposure or habit* or behavior or behaviour or consumption)).mp.) AND (dietary pattern/ or nutritional counseling/ or nutrition education/ or nutrition education.mp. or nutrition misinformation.mp. or ((diet or nutrition or food) adj5 (information or counseling or advise or advice or habit* or behavior or behaviour)).mp.) 800729 hits (social media), 61711 hits (nutrition) Another example from MEDLINE, April 28th: Search for: 4 and 8 and 11 and 12
--	--	--

		Results: 1 Database: Ovid MEDLINE(R) and Epub Ahead of Print, In-Process, In-Data-Review & Other NonIndexed Citations, Daily and Versions <1946 to March 14, 2022> Search Strategy: ----- 1 consumer health information/ (4227) 2 health literacy/ (7781) 3 ((health or nutrition or diet* or medic*) adj information).ti,ab,kw. (39280) 4 or/1-3 (48473) 5 internet/ or social media/ (89510) 6 digital media.ti,ab,kw. (946) 7 ((digital or web or internet or online) adj media).ti,ab,kw. (1319) 8 or/5-7 (90328) 9 Young Adult/ (983772) 10 (young adults or young or early adulthood or early twenties or aged 20).ti,ab,kw. (552717) 11 9 or 10 (1442074) 12 exp "Scandinavian and Nordic Countries"/ (215430) 13 4 and 8 and 11 and 12 (28) The search strategy involved an examination of search terms, an analysis of text words in the title and abstract of retrieved papers, and an analysis of index terms used to describe the articles. A second systematic search was conducted using all identified keywords and index terms, which led to the identification of Jia et al.'s (2021) paper and breakthrough terms such as "Online Health Information Seeking Behavior," and later "young adults". To identify additional studies concerning young people that were not too young, too old or a combination of these factors, both citation pearl and a trial-and-error approach was used. The traditional search string used in reviews for seniors was also adopted, and the same strategy was applied to identify young adults using terms such as "young adults," "aged
--	--	---

		18," and "aged 20." Specific age-related terms were included to ensure that the search results were relevant to the population of interest and to avoid including papers that focus on other age groups. It is a common mistake to include papers including the specific selection AND other cohorts. To ensure research integrity, we wanted to stick to the pre-determined cohort. The Arksey and O'Malley (2005) framework was followed, and papers without a defined cohort or age limitations were excluded to ensure research integrity. This approach was deemed appropriate for the review's scope. References: O'Brien H. L., Dickinson R., Askin N. A scoping review of individual differences in information seeking behavior and retrieval research between 2000 and 2015, Library & Information Science Research, Volume 39, Issue 3, 2017, Pages 244-254, https://doi.org/10.1016/j.lisr.2017.07.007 Jia X, Pang Y, Liu LS. Online Health Information Seeking Behavior: A Systematic Review. Healthcare. 2021; 9(12):1740. https://doi.org/10.3390/healthcare9121740 Percheski, C.& Hargittai, E. (2011) Health Information-Seeking in the Digital Age, Journal of American College Health, 59:5, 379-386, https://doi.org/10.1080/07448481.2010.513406 These changes were made:  - Introduction: Page 1, line 22 - Introduction: Page 2-3, line 65 – 70. - Methods and analysis: Page 6, Table 2 – “Sample” - Methods and analysis: Inclusion criteria nr. 2, page 8, line 238 - 229
11	The final two paragraphs of the Stage 3 section repeat information e.g., the process	Thank you for your valuable suggestions. We have taken them into consideration and carefully reviewed the Preferred Reporting Items for Systematic Reviews and Meta-Analyses extension for Scoping Reviews (PRISMA-ScR) Checklist, specifically items 8-10. Based on our analysis, we have
	of title and abstract screening, full text screening.	determined that the paragraphs in question should remain in the manuscript. We appreciate your input.

12	Expected limitations and strengths of review finding: In this section the authors need to emphasise the implications of the review. How will the review inform strategies to increase the accuracy and reliability of online health information? Be specific.	Thank you for your valuable feedback. We appreciate your interest in our scoping review and understand the importance of addressing issues related to online health information accuracy and reliability. It should be noted that the implications of the scoping review on health information seeking behavior among young adults will be addressed in the final article. This is a standard protocol in academic writing, a transparent recipe of a study to come, where the implications of the study are discussed in the conclusion section of the final paper. However, we would like to clarify that the primary focus of our review is to identify and map the existing literature on this topic, rather than to provide specific recommendations for strategies to increase accuracy and reliability. That is part of the nature to scoping review: to map knowledge gaps and facilitating further research avoiding research waste. The aim is to provide a comprehensive overview of the findings and their potential impact on the field, as well as to suggest future research directions. Subsequently, we acknowledge that our review may have implications for future research and policy development in this area. We will ensure that we discuss these implications in the final paper, including any potential areas for future research and the potential impact of our findings on policy and practice. These changes were made:  - Abstract: Page 1, line 23 - 24
		 - Abstract: Page 1, line 29 - 33 - Ethics and dissemination: Page 2, line 43 – 47 - Strengths and Limitations: Bullet point 2 and 4 in the section has been merged into one, page 2, line 51 and 53. - Methods and analysis: Page 4, line 119 – 122 - Methods and analysis: Page 4, line 125 – 127

VERSION 2 – REVIEW

REVIEWER	McMullan, Ryan Macquarie University Faculty of Medicine and Health Sciences, Australian Institute of Health Innovation
REVIEW RETURNED	15-Jan-2024
GENERAL COMMENTS	The authors have done a good job addressing reviewer feedback. I have no further queries. There are typographical and grammatical errors throughout the manuscript that will need to be corrected before publication.